# Hepatic Cancer Stem Cells: Molecular Mechanisms, Therapeutic Implications, and Circulating Biomarkers

**DOI:** 10.3390/cancers13184550

**Published:** 2021-09-10

**Authors:** Laura Gramantieri, Catia Giovannini, Fabrizia Suzzi, Ilaria Leoni, Francesca Fornari

**Affiliations:** 1Division of Internal Medicine, Hepatobiliary and Immunoallergic Diseases, IRCCS Azienda Ospedaliero-Universitaria di Bologna, 40138 Bologna, Italy; 2Centre for Applied Biomedical Research-CRBA, University of Bologna, St. Orsola Hospital, 40138 Bologna, Italy; catia.giovannini4@unibo.it (C.G.); fabrizia.suzzi3@unibo.it (F.S.); ilaria.leoni2@studio.unibo.it (I.L.); 3Department of Experimental, Diagnostic and Specialty Medicine (DIMES), University of Bologna, 40138 Bologna, Italy; 4Department of Medical and Surgical Sciences, University of Bologna, 40138 Bologna, Italy; 5Department for Life Quality Studies, University of Bologna, 47921 Rimini, Italy

**Keywords:** HCC, treatments, microRNA, stemness, biomarkers, CSCs

## Abstract

**Simple Summary:**

The high heterogeneity of hepatocellular carcinoma (HCC), and the lack of druggable mutations, hamper the identification of unequivocal molecular classifiers and limit the discovery of selective therapeutic treatments. Moreover, the lack of circulating biomarkers guiding the choice of personalized treatments and identifying the occurrence of acquired resistance to treatments still represents an unmet clinical need in HCC. The cancer stem cell (CSC) compartment underlies tumor heterogeneity, disease recurrence, and drug resistance in all cancer types, including HCC. Knowledge of the molecular mechanisms supporting the maintenance and proliferation of CSCs may help to identify novel biomarkers and therapeutic targets to improve and refine HCC management.

**Abstract:**

Hepatocellular carcinoma (HCC) is one of the deadliest cancers. HCC is associated with multiple risk factors and is characterized by a marked tumor heterogeneity that makes its molecular classification difficult to apply in the clinics. The lack of circulating biomarkers for the diagnosis, prognosis, and prediction of response to treatments further undermines the possibility of developing personalized therapies. Accumulating evidence affirms the involvement of cancer stem cells (CSCs) in tumor heterogeneity, recurrence, and drug resistance. Owing to the contribution of CSCs to treatment failure, there is an urgent need to develop novel therapeutic strategies targeting, not only the tumor bulk, but also the CSC subpopulation. Clarification of the molecular mechanisms influencing CSC properties, and the identification of their functional roles in tumor progression, may facilitate the discovery of novel CSC-based therapeutic targets to be used alone, or in combination with current anticancer agents, for the treatment of HCC. Here, we review the driving forces behind the regulation of liver CSCs and their therapeutic implications. Additionally, we provide data on their possible exploitation as prognostic and predictive biomarkers in patients with HCC.

## 1. Introduction

Hepatocellular carcinoma (HCC) is the second leading cause of cancer-related death worldwide, and is one of the most aggressive cancers with an increasing incidence [1]. Despite improvements in screening programs, only 30% of HCC patients are eligible for curative treatments, such as liver transplantation and surgical resection, owing to late diagnosis and compromised liver function [2]. Unfortunately, early diagnostic biomarkers are lacking. In recent years, novel systemic treatments have been approved for advanced-stage HCCs, including first- and second-line molecular targeted agents and immune therapies [3,4]. Current treatments are mainly directed toward the tumor bulk, targeting both the autologous molecular mechanisms driving cancer cell proliferation and invasion, and tumor/microenvironment crosstalk, hitting tumor angiogenesis and immune cell recruitment. In contrast to other types of cancer, neither specific mutations in driver genes, nor druggable mutated genes, guide treatment choices in HCC. In this context, the identification of biomarkers to stratify patients toward optimal therapeutic regimens, or to predict early tumor escape, is still an unmet clinical need.

HCC is a highly heterogeneous tumor with a distinct genetic and molecular background that may respond differently to anticancer treatments. This phenomenon includes spatial heterogeneity within the same HCC nodule and among different tumor nodules in the same patient, or temporal heterogeneity. The latter often occurs after locoregional or targeted treatments, contributing to the acquired resistance of malignant cell clones. In unifocal HCC, different regions of the same tumor can harbor distinct phenotypes and genomic aberrations. In the case of multifocal HCC, tumor heterogeneity is even more pronounced. Indeed, liver nodules can arise from both intrahepatic metastasis and multi-origin lesions, the latter having different genetic and epigenetic alterations conditioning heterogeneous treatment responses [5,6]. Interestingly, the tumor microenvironment displays a lower degree of heterogeneity, aiding a new tumor-immunity-based classification of HCCs that might be helpful for patient management and treatment allocation [7].

The “stem cell theory of cancer” states that there is a minor drug-resistant subpopulation of cancer cells, sharing features with somatic stem cells, that are capable of reproducing themselves (self-renewal) and sustaining tumor mass through asymmetric cell division [8]. These cancer stem cells (CSCs), or tumor-initiating cells (T-ICs), are responsible for tumor heterogeneity and recurrence, as well as for chemotherapy resistance and metastasis. CSCs possess several genomic characteristics favoring the acquisition of aggressive properties, such as enhanced self-renewal ability and adherence-independent growth, facilitating the spread of the tumor and disease relapse. Since CSCs are responsible for tumor heterogeneity and treatment failure in cancers [9], it is of utmost importance to hit not only the tumor mass, but also the small fraction of drug-resistant cells that give rise to tumor initiation and recurrence, in order to achieve complete disease control. In this scenario, the discovery of novel treatment combinations is an urgent clinical need that can improve drug response and avoid, or delay, the onset of acquired resistance.

Here, we review the driving forces behind the regulation of stem cell properties, including oncogenic pathway activation and HCC-specific microRNAs. We also explore the therapeutic implications of targeting the CSC compartment, and evaluate the rationale behind combined strategies in the development of personalized regimens. Finally, we address the issues of the isolation and characterization of circulating CSCs, and their possible use as diagnostic and prognostic tools.

## 2. Molecular Pathways Regulating CSCs in HCC

Several cancer-associated signaling cascades, such as Wnt/β-catenin, NOTCH, Hedgehog, and Oct4, contribute to stem cell-like properties in different tumor types. The roles of the Wnt/β-catenin and NOTCH oncogenic pathways in hepatocarcinogenesis are well-known [10,11]. Here, we describe their influence on the maintenance and expansion of liver cancer stem cells, depicting the molecular mechanisms linked to their aberrant activation and mutual relationships in HCC. Finally, we provide a rationale for focused therapeutic options aiming to switch off these two cancer-related cascades.

### 2.1. Beta-Catenin Pathway

The Wnt pathway is physiologically involved in embryonic development and tissue homeostasis and is pivotal for liver zonation and metabolism. The activation of this signaling cascade occurs at the cell membrane level where Wnt ligands can bind to Frizzled (FZD) receptors triggering two different signaling pathways, namely, the canonical and noncanonical pathways. Specifically, beta-catenin takes part in the activation of the canonical pathway. In the plasma membrane compartment, it regulates cell-cell junctions, while in the nucleus it binds to T-cell factor/lymphoid enhancer factor (TCF/LEF), transcription factors driving the activation of cell cycle promoter and survival-related genes. On the contrary, in the absence of canonical Wnt activation, β-catenin is sequestered and degraded by a complex of proteins composed of Axin1, APC, glycogen synthase 3β (GSK3β), and casein kinase 1 (CK1) [12].

In HCC, the Wnt/β-catenin pathway is the most frequently altered oncogenic pathway due to the activating mutations of the β-catenin gene (CTNNB1, 11–37% of cases), the inactivating mutations of the negative regulators AXIN1 (5–15% of cases) and APC (1–2%) [13], as well as deregulating Wnt receptors, ligands, and antagonists [14]. In particular, comprehensive evaluations of genetic lesions identified β-catenin as one of the most commonly mutated genes in HCC, associating it with chromosome stability, non-HBV infections, and large-size and well-differentiated tumors with a better prognosis [15]. On the other hand, Wnt pathway activation without β-catenin mutations defines a subclass of HCC with high chromosomal instability, an aggressive phenotype, and an association with HBV infections [16].

The Wnt/β-catenin pathway contributes to the induction and maintenance of CSC phenotypes because of the central role that the transcription of downstream genes plays in this process. Adult hepatic progenitor cells, also known as oval cells, are characterized by the expression of two well-known stemness markers, OV6 and epithelial cell adhesion molecule (EpCAM), and by β-catenin signaling activation. In normal livers, this subpopulation of bipotential progenitor cells is responsible for liver regeneration and gives rise to both hepatocytes and cholangiocytes. A subpopulation of less differentiated cells with progenitor-like features and OV6 positivity have been identified in human HCC tissues, showing increased chemoresistance and the ability to form tumors in vivo. Wnt pathway hyperactivation is responsible for their enrichment, and its silencing leads to decreased chemoresistance in OV6+ cells [17]. In addition, the transcriptional activation of the Wnt pathway identified a subset of superpotent triple-positive CSCs (Wnt-activity high EpCAM+ ALDH1+) with the high tumorigenic potential and phenotypical plasticity contributing to a poor prognosis and the tumor heterogeneity of HCC [18].

Besides genomic aberrations, other mechanisms can also contribute to the dysregulation of the Wnt/β-catenin cascade in HCC, improving our understanding of the modulation of CSC properties and offering new paths towards the identification of possible therapeutic candidates, thereby compensating for the undruggable nature of β-catenin itself. Metastasis-associated lung adenocarcinoma transcript 1 (MALAT1) is a long noncoding RNA (lncRNA) with oncogenic properties that is often overexpressed in HCC. Its specific silencing leads to the downregulation of proto-oncogenes (e.g., β-catenin, Myc, and STAT) and epithelial-to-mesenchymal (EMT) markers, and reduces the dye-efflux potential and the aldehyde dehydrogenase 1 (ALDH1) activity of HCC cells. In particular, its interference with Wnt signaling attenuates in vitro tumor sphere formation and in vivo tumor growth, and reduces the subpopulation of CD133+ CD90+ HCC cells, representing an interesting target for CSC-directed treatment [19]. In line with these findings, a 2D-gel approach showed that ALDH1A1 is preferentially expressed in CD133+ HCC cells with respect to CD133- counterparts. In particular, ALDH1+CD133+ cells represent a subpopulation of CSCs with a higher tumorigenic potential [20]. These findings further outline the strong contribution of metabolic changes to the CSC phenotype, as is well-established for Glycine N-methyltransferase (GNMT). Indeed, GNMT exerts its tumor suppressive role by interfering with both the methionine cycle and purine and pyrimidine synthesis [21], and by modulating the expression of cancer-related genes through epigenetic mechanisms. In turn, these complex events lead to the hyperactivation of progenitor OV-6 cells and increase HCC development [22].

Glutaminase 1 (GLS1), a matrix mitochondrial enzyme, is overexpressed in HCC and correlates with stem cell phenotype, advanced clinicopathological features, and poor survival. Its targeting attenuates stemness properties and tumorigenesis by triggering reactive oxygen species (ROS) production and suppressing β-catenin signaling, showing potential as a therapeutic target for CSC eradication in HCC. In particular, glutamine withdrawal or GLS1 inhibition/silencing determines a downregulation of the stemness-associated genes, c-MYC, KLF4, NANOG, OCT4, SOX2, CD13 and CD133, with a concomitant reduction of cell growth and sphere formation. Notably, a positive feedback loop between GLS1 and β-catenin exists, highlighting a mutual regulation of these two players contributing to the maintenance and expansion of the CSC population [23]. GLS1 targeting is of particular interest in cancer, and GLS1 inhibitors are being evaluated in clinical trials in patients with leukemia and solid tumors [24].

Besides Wnt/β-catenin, Akt/mTOR pathway activation is frequently observed in HCC, with 14.4% of the cases activating these dual pathways demonstrating a poorer survival rate. Since the choice of appropriate animal models is particularly relevant in order to assess novel molecularly-based therapeutic strategies [25], the authors chose a hydrodynamic tail vein injection of Akt and β-catenin oncogenes to induce HCC in mice, mirroring this subgroup of human tumors. The establishment of tumor spheres from this HCC animal model helped to enrich a subpopulation of cells with stem/progenitor features, such as CD44 positivity and multi-drug resistance 1 (MDR1) high expression, which is responsible for the dye-efflux in cytometric evaluations. Interestingly, a screening assay for stem cell inhibitors identified JAK/STAT pathway antagonist molecules as the most effective for reducing the cell growth of Akt/β-catenin-driven tumor spheres that display STAT3 overexpression. The use of JAK/STAT inhibitors resulted in the suppression of cell proliferation and a decrease in stem cell markers, supporting the therapeutic effectiveness of JAK/STAT inhibition as a possible therapeutic approach [26]. In summary, Wnt/β-catenin alterations identify a subset of stem cell-like HCCs with a dismal prognosis that might represent the ideal target for a CSC-focused therapeutic intervention.

### 2.2. The NOTCH Pathway

One major barrier to curing cancer is the intratumoral heterogeneity of cancer cells, organized in a hierarchical manner, with a subpopulation of cells characterized by a high capacity for self-renewal. The frequency of CSCs or T-ICs has been reported to be less than 1% in solid tumors [27]. The slow cycling or quiescent state of CSCs is thought to be involved in their resistance to chemotherapy and radiotherapy. Targeting this resistant cell population represents a challenge as stemness-associated factors are shared by both cancer and normal stem cells. Indeed, targeting developmental signaling pathways that specifically regulate the survival of CSCs appears to be an interesting option in HCC.

NOTCH signaling is a cell-cell communication pathway in which NOTCH receptors interact with Jagged or Delta-like ligands on juxtaposed cells [28]. This interaction activates the sequential proteolytic cleavages of the intracellular domain of NOTCH by metalloproteinase (TACE/ADAM17) and γ-secretase [29]. As a result, the NOTCH intracellular domain (NICD) translocates into the nucleus and induces the transcription of target genes belonging to the HES and HEY families. NOTCH signaling emerged as a key regulator of differentiation, cell-fate specification, and stemness. Strong evidence has demonstrated NOTCH signaling activation in CSCs in several malignancies [30]. In the liver, it plays a crucial role in the process of fetal liver stem cell differentiation into hepatocytes, as well as in maintaining the differentiation balance between hepatocytes and cholangiocytes [31,32]. Notably, its inhibition can reverse the malignant phenotype of liver cancer stem cells, driving differentiation into mature hepatocytes [33]. Indeed, the inhibition of NOTCH signaling regains the features of mature hepatocytes, such as albumin production, glucagon synthesis, and urea metabolism via mesenchymal-epithelial transition (MET). These findings suggest that the anticancer effects of NOTCH inhibition may result not only from lowering HCC cell proliferation [34,35], but also from inducing CSC differentiation. In addition, the inhibition of NOTCH signaling induces liver cancer cell death, especially when combined with conventional treatments [36,37,38]. The differentiation of CSCs induced by NOTCH inhibition in HCC is also related to the NOTCH-dependent modulation of the Wnt/β-catenin pathway. Zhang and coauthors reported that 71.8% of HCCs in the Asiatic population have high NOTCH3 expression levels, validating previous evidence found in the Caucasian population [10]. Due to the positive correlation between NOTCH3 and alpha-fetoprotein (AFP), the authors speculated that NOTCH3 positive cells represent a subpopulation of CSCs, and they demonstrated that NOTCH is involved in the maintenance of stemness by regulating the β-catenin pathway [39]. Accordingly, NOTCH and Wnt/β-catenin signaling cascades were shown to play a crucial role in promoting the self-renewal of sphere-forming cells characterized by the expression of CD13, CD90, CD133, and CD24. To further demonstrate the role of NOTCH and Wnt/β-catenin in CSCs, NOTCH and Wnt/ β-catenin pathways were inhibited using γ-secretase (DAPT) and Tankyrase (XAV939) inhibitors, respectively. Stem cell surface markers decreased in cells treated with DAPT or XAV939 and, accordingly, the EMT-associated transcription factors were downregulated [40], as further demonstrated in NOTCH1-silenced HCC cells [41]. However, the decrease in CSC surface markers observed by combining XAV939 and DAPT was the same as that observed by either individual treatment, suggesting a possible crosstalk between these two pathways. Wang and colleagues showed that NOTCH1 is downstream of Wnt/β-catenin signaling and, in turn, NOTCH regulates β-catenin expression to preserve the balance between cell proliferation and the maintenance of the CSC population.

Different mechanisms involved in NOTCH pathway activation have been described in HCC, including the inducible nitric oxide synthase [42], and the hepatocytes nuclear factor-1beta (HNF-1β) that are associated with more aggressive tumors [43]. iNOS promotes NOTCH1 activation through TACE/ADAM17 and induces stemness characteristics in vitro and in vivo, accelerating HCC development. Specifically, iNOS mediates the increase of iRhom2, an intracellular protease critical for TACE/ADAM17 trafficking to the cell surface, driving NOTCH signaling in CD24+ and CD133+ CSCs [44]. In this context, attention has been drawn to designing and developing specific iNOS inhibitors [45]. HNF-1β is expressed in liver progenitor cells and plays an important role in stem cell differentiation into cholangiocytes [46]. Accordingly, HNF-1β overexpression induces the upregulation of liver progenitor cell markers, including CK19, SOX9, and CD133. Moreover, HNF-1β maintains the stemness of liver cancer cells by regulating the NOTCH signaling pathway and, consequently, the EMT-associated genes. As further proof, NOTCH1 silencing in HNF-1β overexpressing cells results in the downregulation of liver progenitor cell markers, highlighting the role of NOTCH1 as a driver of progenitor phenotype [47]. Likewise, LEF1 transcription factor, which is frequently overexpressed in HCC, activates NOTCH1 and NOTCH2 gene transcription through direct binding to their promoter regions. As a result, NOTCH activates downstream targets, including hepatic progenitor markers SOX9 and CK19, and downregulates the hepatic mature marker G6PC, promoting stemness and the poor differentiation of HCC [48]. Even though targeting HNF-1β and LEF1 would represent a new strategy to reverse the malignant phenotype of liver CSCs, interfering with transcription factors is challenging. NOTCH signaling clearly emerged as a prime pathway to be directly or indirectly silenced for the therapeutic targeting of CSCs. Clinical studies have adopted two main approaches to inhibit NOTCH activities, including γ-secretase inhibitors (GSIs), and monoclonal antibodies against NOTCH receptors or ligands. However, functional limitations for these approaches emerged, with adverse effects observed in clinical trials with GSIs [49].

Although limited studies have been carried out, recent findings suggest that the NOTCH pathway regulates the immune checkpoint axis of CSCs in breast cancer. Specific knockdown of different NOTCH receptors showed NOTCH3 as a mediator for PD-L1 overexpression in CSCs and, accordingly, NOTCH3 was found to correlate with high PD-L1 expression [50]. In breast CSCs, NOTCH3 participates in both stemness maintenance and PD-L1 expression. These findings pave the way for a deeper characterization of the CSC-immune microenvironment crosstalk also driven by NOTCH in other cancers. Among these, HCC deserves attention because of the significant role of NOTCH in its development and progression, and because of the promising therapeutic effects of immune checkpoint inhibitors. Possible combinations of immune modulatory and NOTCH inhibitory approaches might be effective tools hitting both the CSC compartment and the tumor bulk.

## 3. MicroRNAs and Stemness in HCC

MicroRNAs (miRNAs) are small noncoding RNAs (≈22 nucleotides) that exert a fine regulation of gene expression by inducing mRNA degradation or translational repression, depending on the extent of base pairing with complementary binding sites located in the 3′-untranslated regions (3′-UTRs) of target genes [51]. Genome-wide studies have reported peculiar miRNA signatures in several cancer types [52], including HCC [53,54], highlighting their deep involvement in tumorigenesis [55]. MiRNAs control multiple biologic functions in HCC, ranging from cell cycle progression [56], to apoptosis [57], invasion [58], and metastasis [59]. MiRNAs also play a critical role in the regulation of CSCs, contributing to the activation of Wnt signaling, and establishing complex feedback loops auto-fueling stemness features [60]. This is the case with miR-5188: the upregulation in HCC is responsible for β-catenin nuclear translocation due to FOXO1 inhibition, preventing its cytoplasmic retention and allowing the transcriptional activation of target genes, such as c-Myc and c-Jun. The latter takes part in a positive feedforward loop, contributing to miR-5188 transcriptional regulation. Moreover, this miR-5188-FOXO1/β-catenin/c-Jun axis is induced by HBV-encoded X protein (HBx), which is an essential component of Wnt signaling and contributes to the poor survival of HCC patients. MiR-5188 overexpression drives the upregulation of the stem- and EMT-associated genes, CD44, Sox2, Oct4, Nanog, ABCG2, ABCB1, Slug, and CCND1 in HCC cells, and is an independent risk factor for HCC [61]. In HCC cells, miR-5188 overexpression increases sphere formation, colony number, and migration and invasion capabilities, and enhances chemotherapy resistance to 5-fluorouracil (5-FU), cisplatin and epirubicin. Three preclinical tools (xenograft, orthotopic, and metastasis mouse models) proved the in vivo role of miR-5188 overexpression in sustaining enhanced tumor growth, intra- and extrahepatic dissemination, and chemoresistance. These findings pave the way for two possible clinical investigations of antagomiR-5188 oligonucleotides. In the first scenario, antagomiR-5188 could be delivered by means of carrier nanoparticles coated with anti-CSC molecules to specifically hit this subpopulation of cancer cells responsible for tumor aggressiveness and recurrence. In the second setting, antagomiR-5188 could be administered during transarterial chemoembolization (TACE) to improve the effectiveness of chemotherapy agents currently used in locoregional treatments. Similarly, the Oct4/miR-1246/Axin2+GSK3β/β-catenin network plays a pivotal role in regulating CSC phenotype and tumorigenesis in HCC. In particular, the nuclear translocation of β-catenin is detected in CD133-enriched HCC cell lines, and miR-1246 resulted in one of the most upregulated miRNAs with respect to CD133- counterparts. Notably, two key components of the β-catenin “destruction complex”, AXIN2 and GSK-3β, are miR-1246 target genes driving β-catenin activation. MiR-1246 knockdown in CD133+ cells decreases their ability to form primary and secondary hepatospheres and invasiveness properties, whereas it increases drug sensitization to both chemotherapies and targeted therapies, and silenced cells fail to give rise to tumors in xenograft mice. MiR-1246 silencing in xenograft models displays increased AXIN2 and GSK3β expression together with decreased β-catenin levels, whereas the orthotopic implantation of miRNA-silenced cells shows a marked reduction of lung metastases, indicating miR-1246 as a pivotal oncomiR mediating tumor initiation and spreading in vivo. Oct4, a self-renewal-associated gene, is responsible for miR-1246 upregulation in CD133+ cells due to direct binding to consensus motifs in its promoter region. MiR-1246 is upregulated in HCC tissue and compared to the surrounding liver. Its high expression is an independent prognostic factor for both overall and disease-free survival [62]. Interestingly, high miR-1246 expression and β-catenin mutations seem to be mutually exclusive events in human HCCs. In this context, this study provides new molecular insights regarding the use of antagomiR-1246, alone or in combination with sorafenib, as a promising CSC-related option in β-catenin WT patients, further emphasizing the need for genetic and molecular classifications of HCC. Notably, not only miRNAs, but also lncRNAs, can promote Wnt/β-catenin pathway activation by acting as competitive endogenous RNAs (ceRNAs). Specifically, the lncRNA DANCR is overexpressed in stem cell-like HCCs and is associated with tumor recurrence and decreased survival. DANCR silencing in HCC cells downregulates the expression of several stemness-associated genes, including CD133, CD90, and EpCAM, and decreases the number and dimension of tumor spheroids. RNA immunoprecipitation revealed an enrichment for *CTNNB1* mRNA. Moreover, bioinformatics analysis displayed the presence of complementary binding sites between DANCR and the 3′UTR of β-catenin. Interestingly, three of these complementary regions are contemporaneously recognized by miR-214, miR-320a, and miR-199a, the first two being bona fide *CTNNB1* suppressor miRNAs. As a proof of concept, an inverse correlation between these three miRNAs and β-catenin mRNA was observed in low DANCR-expressing tumors only, confirming the specificity of DANCR-related ceRNA activity and its competition with miRNAs for the regulation of β-catenin. DANCR interference may spark interest from a therapeutic point of view, as suggested by two HCC animal models where DANCR knockdown with a lentivirus-mediated approach reduced tumor growth, lung metastases, and improved survival. These findings suggest the potential of DANCR as both a prognostic marker and a therapeutic candidate for liver cancer [63]. Since the Wnt/β-catenin pathway is not directly druggable by small molecules, the understanding of its regulatory mechanisms might be relevant for the development of focused CSC-killing anticancer strategies.

Evidence on miRNA-dependent mechanisms driving the CSC phenotype through NOTCH modulation is limited in HCC. However, several miRNAs deregulated in HCC were shown to directly target the NOTCH pathway in other cancer types. For example, miR-200 family downregulation was shown to contribute to stemness features in HCC through the ZEB1 and ZEB2 circuit, as well as by targeting SPAG9, RBBP4, Foxa2, MACC1, and VASH2 [64,65]. In line with this, by repressing miR-200 family members, ZEB1 was shown to enhance NOTCH signaling in cancer cells, contributing to the induction and maintenance of stem cell properties [66]. Similarly, the upregulation of the miR-181 family characterizes a subgroup of highly invasive EpCAM+ HCC cells displaying CSC features. This association was ascribed to the miR-181 targeting of hepatic transcriptional regulators of differentiation, such as caudal type homeobox transcription factor 2 (CDX2), GATA binding protein 6 (GATA6), and nemo-like kinase (NLK) [60]. Interestingly, miR-181 modulation of NOTCH signaling was reported in T-cell acute lymphoblastic leukemia (T-ALL), where miR-181a-1/b-1 was shown to control the strength and threshold of NOTCH activity. Its deletion inhibited NOTCH1-induced T-ALL, pointing to its therapeutic potential [67]. Even though these findings were demonstrated in other cancer types, it appears plausible that the deregulation of these same miRNAs might also play similar roles in the NOTCH-mediated development of the CSC phenotype in HCC. Regarding CSC-related NOTCH pathway regulation in HCC, a brilliant study by Jung et al. describes miR-148a activity in cell differentiation and reports its downregulation as a tumor-promoting event. The authors employed PTEN-null mice that had developed progressive disease, ranging from steatosis to fibrosis, to nonalcoholic steatohepatitis (NASH) and overt HCC. This model is particularly suited to the study of progenitor cell fate due to their accumulation during the different steps of tumorigenesis. MiR-148a in vivo treatment determined decreased tumor growth and degree of malignancy, and showed an increased percentage of hepatocellular and cholangiocellular adenomas with respect to the control group in which HCC nodules were prevalent. In addition, a decrease in progenitor (CD24 and osteopontin) and biliary (KRT19 and SOX9) cell markers, and an increase in hepatocyte cell markers (HNF4A and miR-122), were observed following miR-148a treatment, confirming its positive action on the differentiation of progenitor cells. Furthermore, miR-148a administration before tumor development decreased tumor incidence, suggesting that enhancing progenitor cell differentiation might dampen their self-renewal ability and tumorigenic potential. In this context, a miR-148a mimics-based strategy appears to be a promising differentiation-targeted therapy in HCC. MiR-148a directly targets IκB kinase alpha (IKKa) which, in turn, regulates the NOTCH negative regulator NUMB, determining a reduction of NOTCH signaling as detected by reduced HES1 and HEY1 expression. Since NOTCH2 is the most abundantly expressed NOTCH member in progenitor cells [68], rescue experiments were performed demonstrating that the IKKa/NOTCH pathway mediates miR-148a effects on hepatocyte differentiation. In vivo data with NOTCH inhibitor RO4929097 mirrored findings with miR-148 mimics, further confirming the relevance of the NOTCH pathway in this setting [69].

CD133+ HCC cells display enhanced CSC features, including the ability to grow in cell suspension aggregates (spheroids), the upregulation of stemness-associated (Oct4, KLF4, CD4, EpCAM) and drug-resistant (ABCG2) genes, and the increased resistance to sorafenib. Remarkably, CD133+ cells have a distinct metabolic profile, displaying an increased aerobic glycolysis and extracellular acidification rate (ECAR), coupled with a reduced oxygen consumption rate (OCR) and ATP levels. Interestingly, the silencing of key glycolytic enzymes, lactate dehydrogenase A (LDHA) and pyruvate dehydrogenase kinase 4 (PDK4), impairs stemness properties and increases sorafenib sensitivity. MiR-122 expression is lower in CD133+ HCC cells, and its overexpression inhibits their stem cell-like characteristics through PDK4 targeting and metabolic reprogramming, demonstrating its central role in the regulation of tumor cell metabolism and stemness properties [70]. We also reported a negative correlation between miR-122 and both CD133 and EpCAM stem cell markers in an HCC patient cohort, and demonstrated their negative regulation by miR-122 in HCC cell lines, confirming the active role for miR-122 in stem cell regulation in liver cancer [71]. CD133 expression is associated with a higher recurrence rate and low survival in HCC patients. CD133+ cells are present in small percentages in human HCCs and can be isolated by flow cytometry. Interestingly, CD133+ cells possess increased sphere-formation capabilities and tumorigenic potential in orthotopic xenograft models, displaying an enrichment for T-ICs with respect to CD133- counterparts. In addition, only CD133+ cells isolated from primary tumors can be serially passaged into secondary recipient animals, maintaining the same histological characteristics of the original specimens. MiR-130b is the only miRNA directly associated with CD133 expression in both HCC tissues and cell lines. Its transduction in CD133- cells causes increased cell proliferation, a higher expression of stem-associated genes (β-catenin, NOTCH1, Sox2, Nestin, Bmi-1, and ABCG2), chemotherapy resistance to doxorubicin, and increased spheroid formation and tumorigenicity. The miR-130b biologic effects in HCC are maintained, at least in part, by the direct regulation of tumor protein p53-inducible nuclear protein 1 (TP53INP1) mRNA, and interestingly, CD133 is itself responsible for the positive regulation of miR-130b expression [72].

On the other hand, the liver-abundant miR-192-5p is a tumor suppressor (TS) miRNA strictly associated with cancer stem cells isolated from HCC specimens, resulting in downregulated EpCAM+, CD44+, CD90+, CD133+, and CD24+ CSC-enriched populations compared to negative ones. Its suppression in HCC cell lines increases spheroid formation, CSCs, and pluripotency markers, and reduces metabolism-associated genes typical of normal hepatocytes through poly(A)-binding protein cytoplasmic 4 (PABPC4) targeting. Interestingly, miR-192-5p downregulation in HCC correlates with reduced tumor-free and overall survival, and associates with promoter hypermethylation and p53 mutations, showing an even greater frequency in CD-positive populations [73]. Similarly, we reported the association between miR-30e-3p downregulation and p53 mutations and described the dual behavior of miR-30e-3p in HCC, acting as a TS gene in p53 WT contexts, and as an onco-promoting miRNA in p53 mutated backgrounds. In particular, we demonstrated that miR-30e-3p directly targets EpCAM in HCC cells with high stemness properties, HepG2, and Huh-7 cells. In particular, miR-30e-3p overexpression in Huh-7 cells decreased colony units and sphere number, while miRNA silencing in HepG2 cells produced the opposite effects. On the contrary, miR-30e-3p silencing in EpCAM-negative SNU449 cells reduced invasion and colony formation, probably due to the modulation of other target genes, such as PTEN, activating the AKT pathway and, therefore, behaving as an oncomiR in this experimental setting. In line with in vitro data, an inverse correlation between miR-30e-3p and both AFP and EpCAM mRNA levels was observed in human HCCs, showing an even better correlation in p53-mutated cases, outlining the context-specificity of this miRNA in HCC. MiR-30e-3p expression also plays a pivotal role in the context of sorafenib resistance, as shown by both in vitro and in vivo preclinical models. In particular, miR-30e-3p overexpression induces sorafenib sensitization in TP53 WT HepG2 cells. In agreement, its decreased levels are associated with sorafenib-resistance in tumor nodules from the DEN-induced HCC rat model. Again, the genetic background of HCC, and specifically its TP53 status, is a determinant of direct sorafenib response when a miR-30e-3p replacement strategy is taken into account [74].

HCC animal models reflecting the human disease are useful tools for investigating the role of miRNA in CSC modulation. In particular, c-Met proto-oncogene transgenic (TG) mice developed HCC with a stem-like phenotype through Wnt pathway activation; these tumors are characterized by the overexpression of miRNAs encoded within the Dlk1-Gtl2 imprinted region on mouse chromosome 12qF1 [75]. Similarly, a gene-targeting strategy using an adeno-associated viral [76] vector engineered to site-specifically insert a promoter element into this imprinted region led to liver tumorigenesis in 100% of the mice [77]. Interestingly, both these HCC models overexpress a subset of miRNAs belonging to a mammalian conserved cluster located in the DLK1-DIO3 imprinted locus in human chromosome 14q32.2. This miRNA cluster is upregulated in a subgroup (25–30%) of human HCCs with stem cell properties and correlates with high AFP serum levels and PROM1/CD133 and EpCAM stem-promoting genes [75,78]. Again, half of the upregulated miRNAs in MYC and/or RAS-driven HCC TG mice belonged to this 12qF1 region, and they are upregulated in high AFP-expressing human tumors, showing the highest pro-proliferative potential for miR-494 and miR-495 in HCC cells [79]. We investigated miR-494 involvement in stem cell phenotype and observed a positive correlation with PROM1/CD133 mRNA in two HCC animal models and in human HCCs, demonstrating its positive regulation in miR-494-overexpressing HCC cells and antagomiR-treated xenograft mice. Moreover, miR-494 overexpression induced the upregulation of the stem-associated genes Oct4, Sox2, and ABCG2, enhanced colony formation, and increased sorafenib resistance [78]. A second primate-specific, placental-associated miRNA cluster, located on chromosome 19 (C19MC), is overexpressed in CSC-enriched HCC cell clones and contributes to intratumor heterogeneity. In particular, the side population (SP) isolation method is a flow cytometry technique used to isolate CSCs based on their elevated expression of ATP-binding cassette (ABC) transporters increasing the efflux of certain fluorescent dyes (e.g., Hoechst 33342). The highly invasive cell clone, HCC1, contained the highest SP percentage and upregulated several members of the C19MC together with chromosome X-linked transcripts belonging to the cancer/testis (CT) antigen family. The concurrent overexpression of C19MC miRNA members and CT antigens identifies a subset of HCC patients with decreased overall survival and stem-like features [80]. MiR-589 is upregulated in most HCCs, and its high expression correlates with decreased overall survival and relapse-free survival. Specifically, miR-589 overexpression increases doxorubicin resistance and the mitochondrial membrane potential associated with an increase in antiapoptotic genes (Bcl-2 and Bcl-xL) and a decrease in proapoptotic molecules (BAX and BAK) in preclinical models. Moreover, miR-589 enhances CSC features and improves spheroid formation abilities by increasing CD133+ and SP fractions as well as the stem cell-like genes, Oct4, Nanog, Sox-2, and BMI-1. STAT3 signaling activation mediates miR-589 functions upon direct inhibition of its negative regulators. Intratumor delivery of antagomiR-589 reduces tumor progression and sensitizes cells to doxorubicin in xenograft models. In this scenario, the improved understanding of the mechanisms underlying CSC maintenance may facilitate the development of novel therapeutic miRNA-based strategies to achieve long-term remission and improve advanced HCC prognosis [81]. Similarly, aberrant gene expression sustaining stemness properties in liver tumors may contribute to locoregional treatment failure. In surgically resected patients, miR-125b downregulation is associated with refractoriness to adjuvant TACE with doxorubicin, and the most downregulated cases recorded tumor recurrence within two years after treatment, as well as decreased time to recurrence (TTR) and overall survival. The authors elegantly reported the negative regulation of HIF-1α through a double regulatory mechanism. First, miR-125b directly binds to the internal ribosome entry site (IRES) on HIF1A 5′UTR and negatively regulates its translation and, second, miR-125b targets YB-1, an IRES-dependent translational activator of HIF1A mRNA. Since CD24 is a known transcriptional target of HIF-1α, miR-125b indirectly modulates CSC properties by regulating HIF-1α translation, resulting in increased resistance to doxorubicin and the impaired efficacy of TACE [82].

In summary, here we reported some examples of the importance of HCC-specific miRNA in the survival and maintenance of stem cell properties (Figure 1). The improvement of knowledge regarding the molecular mechanisms mediating miRNA downstream effects in HCC may assist their use as predictive biomarkers and therapeutic targets.

## 4. CSC Targeting and Therapeutic Implications

The persistence of CSCs in treated tumors represents an “Achille’s heel” to be considered when designing anticancer strategies or drug combinations. Hepatic progenitor cells (HPCs) are bipotential liver stem cells located in the canal of Hering that, following specific differentiation signals, give rise to the two hepatic cell lineages, hepatocytes and cholangiocytes. HBx and transforming growth factor beta 1 (TGF-β1) contribute to the malignant transformation of HPCs into hepatic CSCs, and correlate with the stem cell markers EpCAM and CD90 and the poor survival of HCC patients. Zhou and coworkers showed that TGF-β1 treatment, responsible for EMT induction, decreases miR-125b expression in HCC cells. MiR-125b is downregulated in the vast majority of HCCs, and its overexpression attenuates EMT traits, such as cell migration, drug resistance, and stem cell-like phenotype. In particular, miR-125b overexpression sensitizes HCC cells to doxorubicin and sorafenib treatments, and negatively regulates MDR genes, including ABCC1, ABCG2, and ABCB1, contributing to drug efflux from cancer cells. Moreover, miR-125b is downregulated in tumor spheres, and its reinforced expression reduces their diameter, the percentage of EpCAM and CD13-positive cells, as well as tumor initiation in vivo. SMAD2 was reported to be its direct target, and rescue experiments demonstrated its partial involvement in miR-125b-mediated phenotype. MiR-125b mimics in two xenograft models determined a reduction in tumor size and decreased metastatic foci in livers and lungs. Of note, CD13 expression positively correlates with metastatic lung foci, suggesting that miR-125b mimics can inhibit the CSC population in HCC [83]. Interestingly, HBx and TGF-β1 overexpression led to c-Jun N-terminal kinase (JNK)/c-Jun-mediated miR-199a-3p upregulation in the rat-derived hepatic stem cell line LE/6, helping to trigger stem cell-like characteristics in vitro and increased in vivo tumorigenesis [84]. Despite its well-known role as a tumor suppressor miRNA, this study suggests a possible role for miR-199a-3p in the induction and maintenance of the CSC population in HCC. These apparently contradictory findings are in line with the dual cell context-dependent behavior of miRNAs and suggest hitting both the tumor bulk and the CSC compartment as a winning strategy. The studies by Callegari and Varshney [85,86] show the therapeutic potential of miR-199a-3p mimics-based approaches by using different in vivo delivery systems and preclinical tools. In particular, Callegari et al. demonstrated the effectiveness of miR-199a-3p mimics in reducing tumor growth in the miR-221 TG mouse model, and confirmed the regulation of its well-known target genes, mTOR and p21 activated kinase 4 (PAK4), as one of the main molecular mechanisms responsible for its antitumor activity. Notably, miR-199a-3p enhanced expression showed an antitumor effect comparable to that obtained with sorafenib treatment, with no additive or synergic effect for the combined treatment, suggesting that miR-199a-3p mimics work better alone rather than in combination in HCC. On the other hand, Varshney and collaborators established a new formulation for the in vivo delivery of miR-199a-3p by using self-assembled dipeptide nanoparticles and confirmed its anticancer activity by mTOR direct regulation. Interestingly, in stem cell-like HCC subgroups, a specific miR/antagomiR delivery to CSCs could be obtained by using stem-targeted nanoparticles to selectively eradicate the CSC compartment.

NF-κB pathway hyperactivation is frequently observed in cancer, and it contributes to sorafenib resistance in HCC, constituting a possible hit for combined treatments. Strikingly, cytochrome P450 1A2 (CYP1A2) is downregulated in most HCCs and it negatively correlates with NF-κB p65 expression levels. Omeprazole is a CYP1A2 inducer, and its co-administration enhances sorafenib activity in preclinical HCC models by switching off nuclear NF-κB signaling [87]. Whether this combination could reduce sorafenib resistance in patients with HCC needs to be clarified. In addition, NF-κB activation following sorafenib treatment promotes enrichment for T-ICs by mediating the transcriptional upregulation of CD47, and by enhancing stemness properties, such as self-renewal, in vivo tumorigenesis, and invasiveness. A positive correlation between NF-κB and CD47 expression is observed in HCC specimens. Interestingly, CD47 is a “don’t-eat-me signal” expressed by almost all cancer cells, preventing the phagocytic eradication of malignant cells by innate immunity surveillance. Inhibition of CD47 signaling by monoclonal antibodies (mAb) helps macrophages to eliminate cancer cells and reduces tumor growth, leading to a potential curative approach when tumors are treated in their initiation phase [88]. Strikingly, CD47 inhibition by a lentiviral-based strategy or mAbs induces drug sensitization in sorafenib-resistant clones. Anti-CD47 mAb (B6H12) exerts a synergistic effect in patient-derived xenograft (PDX) mice, suggesting this combined regimen as an interesting approach for HCC treatment [89]. In agreement, curcumin treatment leads to a specific inhibition of the NF-κB signaling pathway resulting in a selective targeting of the CSC population and reduced tumor growth [90].

Due to enhanced proliferation in tumors, cell cycle inhibitors constitute a promising therapeutic approach for HCC and some candidates have entered clinical trials [91]. Cyclin-dependent kinase 1 (CDK1) is upregulated in HCC, and its expression correlates with poor overall survival. CDK1 inhibition by the small molecule RO3306 increases sorafenib sensitization by blocking the CDK1/PDK1/β-catenin pathway and the expression of the pluripotency markers Oct4, Sox2, and Nanog, exerting an inhibitory effect on the CSC compartment. In particular, a synergistic antitumor effect was registered on PDX tumor models, which represent a useful tool for preclinical and clinical drug development studies because of their similarity with the tumor of origin in terms of molecular, histologic, and genomic characteristics. In addition, β-catenin inhibition reduced the EMT of cancer cells, also showing antimetastatic potential. CDK1 targeting represents a promising strategy for enhancing sorafenib sensitivity by overcoming resistance and improving the outcome of advanced HCCs [92]. Another study reported the antitumor and synergic effect of palbociclib (PD-0332991) in association with sorafenib. Palbociclib is a selective CDK4/6 inhibitor affecting the cell cycle progression of retinoblastoma-1 (Rb-1) wild type HCC cells and impairing tumor growth in vivo. Notably, palbociclib treatment induces cellular senescence in a large proportion of cancer cells in xenograft mice. Nevertheless, reversible cell cycle arrest is observed in the remaining malignant cells, likely accounting for tumor relapse after treatment withdrawal [93]. We can speculate that palbociclib-resistant cells might represent the CSC subpopulation whose cell cycle is generally arrested, possibly explaining their drug resistance to cell cycle inhibitors.

Chronic liver diseases associated with different etiologic factors (HBV, HCV, alcohol abuse, obesity, and metabolic syndrome) activate a proinflammatory state driven by resident immune cells that secrete chemokines and cytokines to attract further proinflammatory immune cells. This contributes to disease progression, ranging from chronic hepatitis to liver cirrhosis and, finally, to HCC. Repeated necrosis and regeneration cycles cause the accumulation of mutations, driving the malignant transformation of hepatic progenitor cells into CSCs favoring tumor progression. Intriguingly, treatments with sorafenib and DNA intercalators increase the CSC population. On the contrary, mTOR and NOTCH inhibitors (e.g., rapamycin and DAPT) reduce the CD133^+^/EpCAM^+^ enrichment of both the epithelial and mesenchymal-like HCC cells, showing that rapamycin prior to sorafenib administration is the best option able to reduce the formation of tumor spheres. Impressively, interleukin-8 (IL-8) gene upregulation associated with CSC enrichment following sorafenib treatment showed the opposite behavior in the presence of rapamycin or DAPT administration. In line with this, IL-8 inhibition by Reparixin or siRNA molecules sensitized HCC cells to sorafenib by decreasing the CSC compartment. Interestingly, regorafenib also increases the expression of stemness-promoting interleukins (IL-8, IL-11, IL-1b), leading to the enrichment of stem cell-like populations. This study highlights adjuvant therapies against CSCs as a possible strategy for boosting the targeted therapy response in HCC patients [94]. Two first-line treatments, sorafenib and lenvatinib, with different kinase-activity profiles, are currently available for advanced-stage HCCs [95] with no biomarker helping patients’ allocation. Interestingly, lenvatinib, but not sorafenib, decreases the CD133^+^/CD44^+^ CSC population in HCC preclinical models by blocking FGFR1-3 signaling, which is indeed one of the main differences between these two drugs. In agreement, a positive correlation between CD133^+^/CD44^+^ and FGFR1-3 or FGF2 was found in an HCC cohort from The Cancer Genome Atlas [96]. This study shed light on the possible subclasses of HCC patients that could benefit from lenvatinib treatment as a first-line agent, possibly avoiding sorafenib resistance because of CSC induction.

Immunotherapy represents another attractive strategy for hitting the CSC compartment to complement conventional therapies that are mainly directed toward the tumor bulk without specifically killing quiescent malignant cells. Annexin A3 (ANXA3) is highly expressed in CD133+ cells, regulating their proliferation, expansion, and immunophenotype. In particular, ANXA3 overexpression increases EpCAM and CD90 and CD44 levels, and transactivates the HIF1A gene, leading to CD133, NOTCH1, and NOTCH2 upregulation. Pan et al. demonstrated that ANXA3-transfected dendritic cells activate cytotoxic T cells preferentially directed against CD133+ HCC cells [97]. Notably, the number of intratumoral cytotoxic T cells is not increased in CD133^+^ ANXA3^high^ HCCs, indicating that infiltrating T cells are mainly exhausted, as brilliantly confirmed by a single-cell analysis depicting the ecosystem and immunophenotype of early relapse HCCs [98]. In this scenario, the immune modulation of CD8+ T cells represents an interesting paradigm for eradicating liver tumors [3]. To sum up, here we reported some molecular mechanisms sustaining resistance to the treatment of CSCs. These data emphasize the need for combinatorial approaches directed, not only to the tumor bulk and microenvironment, but also to the CSC component (Figure 2). Of utmost importance, knowledge of the molecular dynamics regulating CSC-mediated tumor evolution could help to identify novel therapeutic targets, as well as biomarkers, for patients’ allocation to treatments. Indeed, to date, the clinical translation of cancer stem cell-based diagnostic and therapeutic markers is still very limited in the field of HCC. The majority of the studies registered so far on the ClinicalTrials.gov platform do not report related results. Most of the studies are aimed at the detection and sorting of circulating cancer stem cells from peripheral blood in HCC patients. In particular, these studies: investigate the prognostic value of circulating CSCs on postoperative recurrence and metastasis (NCT02727673) or; assess biomarkers associated with hepatoblastoma subtypes and cancer stages (NCT01336881) or; try to identify correlations with functional imaging and clinical course in the setting of antiangiogenic treatments (NCT01507740).

## 5. Circulating Cancer Stem Cells

Circulating tumor cells (CTCs) and circulating cancer stem cells (CCSCs) contribute to metastasis and were initially proposed as diagnostic and prognostic biomarkers in breast, prostate, and colon cancers [99,100,101,102]. CTCs and CCSCs are rare in the bloodstream, and this increases the difficulties encountered in the development of analytical methods for their enumeration and characterization. Indeed, both the number and the immune phenotype of CTCs and CCSCs are relevant in defining their biological significance. Even more relevant is the separation of CTCs from CCSCs by specific biomarkers. Despite their biologic importance, a univocal definition of these two circulating cell populations is still a matter for research. Indeed, heterogeneous panels of biomarkers have been used to identify either CTCs or CCSCs, and their distinction is sometimes unclear in clinical studies. Different approaches have been used to enumerate these circulating cells. One of the most common is based on immunomagnetic capture using specific antibodies (mainly against EpCAM), followed by immunofluorescence characterization with anticytokeratin antibodies to confirm the epithelial origin, and anti-CD45 to rule out the presence of leucocytes. Biomarkers able to recognize CCSCs were used based on preliminary studies on tissue CSCs. An FDA-approved system (CELLSEARCH^®^ Circulating Tumor Cell Kit–Menarini Silicon Biosystems Inc, Bologna, Italy) utilizes EpCAM and cytokeratins (CK8, CK18, and CK19) in CD45 negative circulating cells. By using this technique, CTCs were identified as indicators of poor prognosis after surgery as well as after TACE of HCC [103,104]. EpCAM is expressed in premalignant hepatic tissues and in a subgroup of HCCs. Its expression is associated with a distinct molecular signature with the features of hepatic progenitor cells, such as cytokeratin 19, c-Kit, and Wnt/β-catenin signaling. However, EpCAM is not expressed in the majority of HCCs, being identified in less than 50% of cases [105]. In addition, loss of EpCAM expression characterizes cells undergoing EMT, which is a common trait of metastatic cells [106]. Thus, identifying the optimal biomarker panel to capture CTCs or CCSCs is still an open issue and should be tailored for each specific tumor type. Other analytical approaches include the so-called “label-free” methods, such as density-based gradient centrifugation, size-based filtration, and microfluidic-based technology that captures circulating cells based on their size and deformability properties (Parsortix™ Cell Separation System; ANGLE North America, Inc., King of Prussia, PA, USA) [107], avoiding cell loss since they do not use preliminary capture steps. Alternative approaches are based on the depletion of leucocytes followed by the reverse transcription (RT)-PCR detection of epithelial and cancer stem cell transcripts, or microfluidic platforms enabling on-chip CTC isolation and identification offering the possibility of culturing cells. While the first approach (immunomagnetic capture followed by immunofluorescence characterization) mainly enumerates CTCs, the RT-PCR detection of epithelial transcripts in CTCs after leucocyte depletion seems to be more sensitive and predictive of prognosis in breast, colon, and metastatic prostate cancers [108,109,110]. Both these assays are suboptimal for sensitivity and specificity. Despite technical and biological concerns, the contribution of CCSCs to the diagnostic and prognostic perspective is gaining attention. Remarkably, the enumeration of CTCs and CCSCs needs to be corroborated by their characterization, which ultimately provides clinically relevant prognostic and predictive information. In the last two decades, several studies aimed to characterize and prognosticate HCC patients based on noninvasive biomarkers. In this perspective, CCSCs deriving from HCC were sought in the blood of patients treated by surgical hepatectomy as a possible predictor of disease recurrence. Fan et al. used flow cytometry to analyze the peripheral blood of 82 HCC patients before hepatectomy, detecting the CCSCs identified as CD45- CD90+ CD44+ cells [111]. Patients experiencing HCC recurrence demonstrated a higher number of CCSCs than patients without recurrence. A higher number of CCSCs predicted both intra- and extra-hepatic recurrence as well as a lower 2-year recurrence-free and overall survival rate. These data make CCSCs a novel parameter worthy of investigation as a noninvasive biomarker in HCC patients, not only in the surgical setting, but also in the postsurgical work-up, tailored to upgrade the surveillance and treatments for patients with a higher risk of early HCC metastasis.

CD133 is one of the most credited biomarkers used to detect CCSCs. Ma and collaborators [112] isolated a CD133+ population from Huh7 PLC8024 HCC-derived cell lines, and from the HepG2 hepatoblastoma cell line, and they showed that colony-forming efficiency, proliferation, and the ability to form tumors in vivo were higher in CD133+ populations. CD133+ cells display progenitor cell characteristics, such as the expression of “stemness” genes, and the ability to self-renew and to differentiate into nonhepatocyte-like lineages. These data support CD133 as an ideal biomarker recognizing tissue CSCs in HCC. CD133 is a transmembrane hematopoietic stem cell antigen expressed in human fetal liver and liver-repairing tissues and is correlated with tumorigenicity in HCC, as well as in other malignancies. Indeed, CD133 is expressed on a small cell fraction of human HCC-derived cell lines and primary HCC tissues. Remarkably, CD133-positive cells demonstrate high tumorigenicity and clonogenicity compared with CD133-negative ones [113]. Uncovering the molecular mechanisms driving CD133 expression may help identify combined strategies against the CSC contribution to cancer progression and resistance to treatments. At the tissue level, Ma et al. [72] determined the percentage of CD133+ cells in the HCC nodule as ranging from 1.3% to 13.6% by using either flow cytometry or histological analysis. CD133+ cells were localized in the tumor bulk and they were responsible for the maintenance and growth of HCC. Conversely, CD133 expression could not be detected in nontumor liver tissues. These authors validated CD133 as a marker for liver tumor-initiating cells in human HCC, with a prognostic significance. As mentioned above, they also identified miR-130b as a regulator of the growth and self-renewal of CD133+ T-ICs by targeting TP53INP1 mRNA. A negative correlation between both CD133 and miR-130b and TP53INP1 is present in HCC cell lines and patient-isolated HCC cells. Silencing and rescue experiments proved the decisive role of TP53INP1 in spheroid formation and in vivo tumorigenesis, demonstrating its targeting by miR-130b as a leading event promoting CSC phenotype. Since the reactivation of CSCs is critical for tumor recurrence and drug resistance, the understanding of miRNA-based molecular mechanisms underlying the regulation of this cell population may provide more effective cancer strategies against HCC.

Besides CD133, CD44 also holds promise as a putative tissue biomarker expressed by a liver cell population with higher metastatic potential [114,115]. Hou and collaborators [116] showed that the CD133+CD44/high pattern identifies the fraction of tumor cells responsible for hematogenous metastasis in HCC. This pattern was associated with portal vein invasion. CD133+ or CD44+ HCC cells demonstrated higher clonogenic growth and vascular invasion compared to their negative counterparts and were associated with intrahepatic and lung metastasis development in nude mice. These authors suggested CD133+CD44/high tumor cells as a predictive biomarker of hematogenous metastasis, and a possible target for reducing HCC metastatization. To characterize circulating CD133+ cells in HCC patients, Zekri et al. [117] analyzed the expression of 13 miRNAs in purified CD133-positive cells separated from the peripheral blood of healthy volunteers and from patients with HCV-related chronic liver diseases (CHC), cirrhosis (LC), and HCC. Three panels of partially overlapping miRNAs were deregulated in HCC versus CHC patients (miR-602, miR-122, miR-181b, miR-125a-5p downregulation, and miR-192 upregulation), in HCC versus the LC group (miR-199a-3p, miR-192, miR-122, miR-181b, miR-224, miR-125a-5p, miR-885-5p upregulation, and miR-22 downregulation), and in CD133+ cells from the HCC group compared to CHC and LC patients (miR-192, miR-122, miR-181b and miR-125a-5p upregulation). This last four-miRNA panel was suggested to characterize circulating CD133+ cells in patients with HCV-related HCC and provide the basis for detecting CCSCs by using an RT-PCR-based approach. However, this assay does not allow single cell analyses. Remarkably, CD133 is also expressed by endothelial progenitor cells [118] and PCR-based assays cannot determine the cellular origin of CD133+ cells (endothelial progenitor cells or CCSCs). From this perspective, approaches coupling the negative selection of hematopoietic cells followed by the analysis of CCSC-associated antigens seem to be more specific.

Xu and collaborators [119] successfully developed a magnetic separation method of circulating HCC cells by binding an asialoglycoprotein receptor (ASGPR) ligand to biotinylated asialofetuin, followed by labelling with antibiotin antibody-coated magnetic beads to obtain cell isolation. Circulating tumor cells were detected in 81% of HCC patients. By adopting an anti-ASGPR antibody instead of an ASGPR ligand, Li et al. obtained a successful CTC detection in 89% of HCC patients [120]. Other ligands expressed by HCC-derived CTCs used for isolation procedures include Glypican-3 (GPC3) [121], Vimentin [122], or panels of cell-surface markers such as ASGPR, GPC3, and EpCAM [123].

Circulating CSCs are very rare, and their identification may be challenging in laboratories lacking the required technologies, so other biomarkers associated with their presence have been investigated. Among these, recent studies focused on circulating extracellular vesicles (ECVs). Due to their cargo of nucleic acids, lipids, and proteins, ECVs are involved in cancer cell crosstalk with both the tumor microenvironment and immune system cell populations. Brocco and collaborators [124] investigated peripheral blood ECVs by flow cytometry, evaluating their number and immunophenotypic characteristics in cancer patients. Besides confirming the higher number of ECVs in metastatic and locally advanced nonhematological cancer patients, these authors observed a higher concentration of circulating ECVs originating from endothelial (CD31+) and tumor cancer stem cells (CD133+ CD326-). In addition, higher levels of CD133+ CD326- ECVs are correlated with poorer overall survival. These data are in line with the dismal prognostic role of CD133 expression in cancer tissues [125], and support the future exploitation of CD133+ CD326- ECVs as diagnostic and prognostic biomarkers to be evaluated in defined cancer types and treatment settings. In this context, we observed a p53-dependent secretion of miR-30e-3p in exosomes from HCC cell lines and reported an association between higher miR-30e-3p circulating levels and sorafenib resistance in advanced HCC patients. Even though still preliminary, our findings suggest this stem cell-associated miRNA as a candidate for use in the the early prediction of tumor escape in liquid biopsies from treated HCC patients [71].

To summarize, CTCs and CCSCs are thought to play a crucial role in metastasis and resistance to treatments. Their characterization and quantitative assessment can also provide relevant information in the setting of response to treatments in advanced HCC. Even though a panel of biomarkers for the identification and quantification of tissue and circulating CSCs is yet to be defined, studies in progress aim at identifying informative biomarkers to optimize analytical procedures and evaluate their clinical validity.

## 6. Concluding Remarks

Divergent lines of evidence accredit either progenitor cells or hepatocytes with the ability to undergo transdifferentiation in the liver cell population, giving rise to HCC [126]. Accordingly, the phenotypic markers of hepatic CSCs are still not fully determined, and the existence of different CSCs subpopulations with a dynamic interconversion is still a matter of debate. Consequently, it is still unknown whether the cancer stem markers recognized in other cancer types can also be applied to HCC. It is likely that the choice of different isolation procedures and heterogeneous cancer stem cells markers allow the selection of CSC-enriched subpopulations, even though there is high heterogeneity among studies. When looking for CCSCs and CTCs, these issues become even more relevant because the analytical procedures and biomarkers still need to be univocally defined and optimized. Some phenotypic molecules such as EpCAM, CD133, CD90, CD44, CD133, and panels of miRNAs, were used in clinical studies to identify the presence and the abundance of CTCs and CCSCs in HCC patients. However, much remains to be done in order to: (1) validate these preliminary data; (2) establish a shared panel of biomarkers; (3) characterize the subpopulations identified by specific biomarker panels; and (4) assess their clinical usefulness in HCC patients. In this context, knowledge of molecular events driving the expression of each biomarker, and their functional relevance for determining the “stemness phenotype”, is of paramount importance in order to accurately select the biomarkers to be further investigated in specific clinical settings. Preliminary studies show that the presence of a larger number of CCSCs defines subgroups of HCC patients with a worse prognosis. Indeed, while the contribution of cancer stem cells to the tumor bulk is believed to be marginal, their presence is nevertheless associated with resistance to therapy. The circulating fraction of CSCs derived from primary or metastatic lesions migrates into circulation and is considered the “seed” of tumor metastasis. In concept, CTCs and CCSCs represent the most informative component of a liquid biopsy since they are supposed to be live tumor cells carrying a comprehensive bulk of information about the most aggressive components of the tumor of origin [127]. Since no consensus exists on analytical techniques and biomarker panels to enumerate and characterize CCSCs or CTCs in different studies conducted in HCC patients, conclusion on the clinical meanings of CCSC or CTCs still seems preliminary. Notwithstanding, data published so far hold promise for a possible incorporation of liquid biopsy, with particular regard to CCSCs, in the characterization of HCCs, especially in those cases in which tumor biopsy is not suitable.

Another crucial field in which cancer stem cells are thought to play a pivotal role is the emergence of resistant clones during the course of targeted treatments, such as sorafenib [128]. Indeed, sorafenib-acquired resistant clones are enriched in progenitor cell-like signatures. These resistant cells also demonstrated an increased capability for sphere formation and tumorigenic potential. The enrichment of CSCs/T-ICs during the development of drug resistance and disease progression under sorafenib treatment indicates this line of research as a promising field, not only as a source of prognostic markers, but also as a target for novel treatments. Thus, efforts should be addressed to the standardization of reproducible analytical procedures, biomarkers for HCC-specific CTCs and CCSCs, the establishment of cut-off values, and the definition of subgroups of HCC patients to be tested. Even though technical and clinical efforts are mandatory before any clinical use of these findings, the preliminary results show great promise in diagnostic and prognostic settings, as well as in the prediction of response to treatments and their possible exploitation as therapeutic targets.

## Figures and Tables

**Figure 1 cancers-13-04550-f001:**
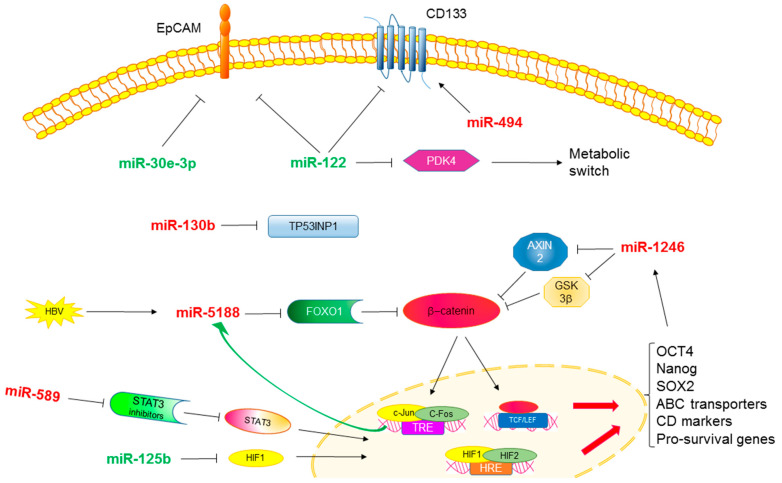
MicroRNAs and CSCs in HCC. Regulation of stemness molecules and pathways by tumor-specific miRNAs, leading to transcriptional activation of pro-survival genes and stemness-associated genes in HCC.

**Figure 2 cancers-13-04550-f002:**
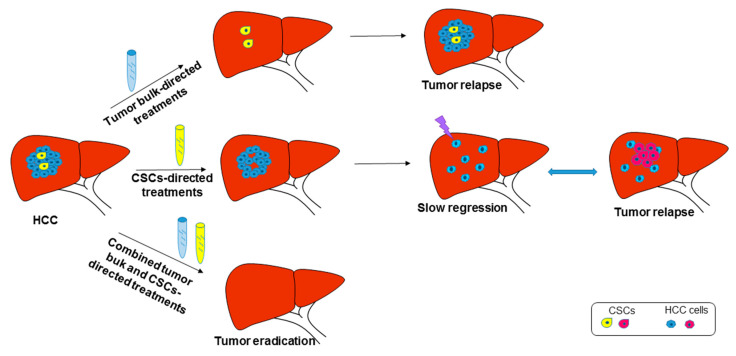
Tumor bulk and CSC-based combined treatments of HCC. Top panel: tumor bulk-directed treatments might lead to tumor relapse due to CSC reactivation. Central panel: CSC-directed treatments might lead to tumor relapse due to the acquisition of further DNA mutations and the possible transdifferentiation of tumor cells. Bottom panel: increased efficacy of combined treatments leading to tumor eradication.

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
