# Peer review of "Hepatic Cancer Stem Cells: Molecular Mechanisms, Therapeutic Implications, and Circulating Biomarkers"

_cancers, 2021, doi:10.3390/cancers13184550_

Round 1

Reviewer 1 Report

This review article provides a good overview of the major known mechanisms of hepatocarcinogenesis. The mechanisms and relationships described here are relevant and highlight potential targets for future therapeutic approaches and research priorities. The aspects described here are of relevance and interest to the relevant scientific field.

This successful review article is understandable and well-structured and reflects an intensive examination of the current literature. The authors' conclusions are described and discussed consistently with the relevant literature. The main signaling pathways and mechanisms are summarized in an appealing graphic. The amendments elaborate on the focus of the review more specifically and add to the paper accordingly. I fully recommend to accept the current version for publication. 

Author Response

We are grateful to Reviewer 1 whose suggestions led to an improvement of our paper.

Reviewer 2 Report

The article is well written with regular flow and smooth transition from one topic to another. The review has meticulously covered the extent of research with recent articles and research studies. Figures are well descriptive, and authors have collected interesting scientific information and have thoroughly reviewed the literature which is imperative for a review article

Though I have minor comments for the authors. 

  • The authors are suggested to include and talk about folate metabolizing enzymes like ALDH1L1 and GNMT in HCC. Folate has been studies extensively for its role in HCC for decades and ALDH family is a potent marker of stem-ness.
  • Human protein atlas and animals tissue atlas should be thoroughly reviewed.

Author Response

As requested by the Reviewer, we have added some data on ALDH1 and GNMT deregulated expression in the liver cancer stem cells compartment. In particular, we have added three new References (20-22) in the ‘Beta-catenin pathway’ chapter highlighting the strong contribution of metabolic changes to CSC phenotype. However, since this topic is not within our specific research fields, we have decided not to include it in the original version of our manuscript and not to examine it in depth. 

For each study mentioned in this review, we extensively described in vitro and in vivo models. We also reported the diagnostic/prognostic or therapeutic significance of each molecular pathway described in our manuscript in order to improve HCC management and/or response to therapy. We think that the revision of human protein atlas and animal tissue atlas is beyond the aim of this review.

As previously requested, a native English speaker has performed an extensive English revision.

This manuscript is a resubmission of an earlier submission. The following is a list of the peer review reports and author responses from that submission.

Round 1

Reviewer 1 Report

The review manuscript from Gramantieri et al. is summarizing in a detailed way novel findings indicating that HCC cancer stem cells are the main population within a tumor responsible for therapeutic failure. To explain the molecular mechanisms responsible for CSC resistance they are focusing on the role of Wnt/ Beta-catenin and Notch pathways and microRNAs. microRNAs are the RNA family predicted to have a high clinical impact and prognostic potential. Unfortunately, the clinical relevance and the summary of data from clinical studies are missing in the manuscript. This would be necessary to add.

Comments
1. Majority of this review just roughly described the renowned background of biological effect and therapeutic implications of hepatic Cancer stem cells. The role/mechanisms of microRNAs involved in hepatic Cancer stem cells regulation through Wnt/ Beta-catenin and Notch pathways should be included in the section of “MicroRNAs and stemness in HCC”. Additionally, CSCs related MicroRNAs implicated in therapeutic implications of HCC should be discussed here.

  1. The part of “CSCs targeting and therapeutic implications” and “Circulating cancer stem cells” is short on mechanism investigation. The quality of this article could be improved by extending the content regarding the effects and mechanisms of microRNAs in determining the expression of CSCs regulators or markers and the sensitivity after cancer therapies).

    1. This manuscript is not well written in English. The context throughout the article should be corrected by a native-speaking English person.

Author Response

Reviewer #1

The review manuscript from Gramantieri et al. is summarizing in a detailed way novel findings indicating that HCC cancer stem cells are the main population within a tumor responsible for therapeutic failure. To explain the molecular mechanisms responsible for CSC resistance they are focusing on the role of Wnt/ Beta-catenin and Notch pathways and microRNAs. microRNAs are the RNA family predicted to have a high clinical impact and prognostic potential. Unfortunately, the clinical relevance and the summary of data from clinical studies are missing in the manuscript. This would be necessary to add.

Comments

  1. Majority of this review just roughly described the renowned background of biological effect and therapeutic implications of hepatic Cancer stem cells. The role/mechanisms of microRNAsinvolved in hepatic Cancer stem cells regulation through Wnt/ Beta-catenin and Notch pathways should be included in the section of “MicroRNAs and stemness in HCC”. Additionally, CSCs related MicroRNAs implicated in therapeutic implications of HCC should be discussed here.

R: As requested, we have moved REF #20 of the original manuscript (now REF #60 of the revised version) from the Wnt/Beta-catenin section to the “MicroRNAs and stemness in HCC” section, which has been expanded with further mechanistic details. In addition, we have added new studies on Notch pathway regulation by cancer-deregulated miRNAs and described their influence on CSC phenotype, emphasizing their therapeutic implications for HCC (REFs #61-66).

Moreover, we have implemented the description of molecular mechanisms and therapeutic implications of CSCs-related microRNAs for REFs # 60, 58, 59, 69, 71 (old REFs #20, #59, #60, #63, #65) in the “MicroRNAs and stemness in HCC” section. We thank the Reviewer for this suggestion, which allowed us to further explain the possible therapeutic implications of targeting CSCs by miRNA-mediated approaches.

  1. The part of “CSCs targeting and therapeutic implications” and “Circulating cancer stem cells” is short on mechanism investigation. The quality of this article could be improved by extending the content regarding the effects and mechanisms of microRNAs in determining the expression of CSCs regulators or markers and the sensitivity after cancer therapies).

R: The description of molecular mechanisms of microRNAs and their therapeutic implications was added for REFs #82, 83 (old REFs #75 and 76) and newly added REF #80 in the CSCs targeting and therapeutic implications” chapter and for REFs #69 and 71 (old REFs #63 and 65) in the “Circulating cancer stem cells” chapter.

The clinical relevance and the summary of data from clinical studies have been added at the end of the “CSCs targeting and therapeutic implicationschapter. To our knowledge, the clinical translation of cancer stem cell-based diagnostic and therapeutic markers is still very limited in the field of HCC. The available studies are mainly oriented towards the investigation of (1) circulating CSC potential on postoperative recurrence and metastases, (2) biomarkers association with hepatoblastoma subtypes and cancer stages and (3) correlations with functional imaging and clinical course in patients treated with antiangiogenic agents.

  1. This manuscript is not well written in English. The context throughout the article should be corrected by a native-speaking English person.

R: A native English speaker has performed an extensive revision of our manuscript.

Reviewer 2 Report

Your review article “Hepatic cancer stem cells: molecular mechanisms, therapeutic implications and circulating biomarkers” is an excellent overview on the role of cancer stem cells in failed treatments of hepatocellular carcinoma. You also review the importance of targeting the cancer stem cells in combined treatment strategies.

English language and style are fine/minor spell check required 

Author Response

Reviewer #2

Your review article “Hepatic cancer stem cells: molecular mechanisms, therapeutic implications and circulating biomarkers” is an excellent overview on the role of cancer stem cells in failed treatments of hepatocellular carcinoma. You also review the importance of targeting the cancer stem cells in combined treatment strategies.

English language and style are fine/minor spell check required 

R: A native English speaker has performed an extensive revision of our manuscript. We thank the Reviewer for this suggestion.

Reviewer 3 Report

HCC is a very complex tumor that is rarely detected in the early stages and is difficult to treat in the advanced stages. Beside many different mechanisms and signaling pathways involved in the progression of the tumor, the authors also describe the impact of HCC-specific microRNAs in this review.
The authors also address the role of cancer stem cells (CSCs) or tumor initiating cells (T-ICs), which are described as drivers of tumor heterogeneity and recurrence, as well as chemotherapy resistance and metastasis and therefore demonstrate potential diagnostic, prognostic and therapeutic tools to improve HCC therapy strategies.
HCC is, as already mentioned, a very complex tumor entity, is unfortunately often recognized at an advanced stage, which is why successful therapy is difficult and complex. The recurrence rate is very high depending on the genesis and also complicates the therapeutic success in the disease. The
mechanisms and correlations described here are therefore relevant and highlight potential targets for future therapeutic approaches and research points. The aspects described here are of importance and interest for the corresponding scientific field.
The article is understandable and well-constructed, and reflects an intensive examination of the current literature. The authors’ conclusions are consistently described and discussed by relevant literature. The essential signaling pathways and mechanisms have been summarized in an appealing figure.

Author Response

Reviewer #3

HCC is a very complex tumor that is rarely detected in the early stages and is difficult to treat in the advanced stages. Beside many different mechanisms and signaling pathways involved in the progression of the tumor, the authors also describe the impact of HCC-specific microRNAs in this review.
The authors also address the role of cancer stem cells (CSCs) or tumor initiating cells (T-ICs), which are described as drivers of tumor heterogeneity and recurrence, as well as chemotherapy resistance and metastasis and therefore demonstrate potential diagnostic, prognostic and therapeutic tools to improve HCC therapy strategies.
HCC is, as already mentioned, a very complex tumor entity, is unfortunately often recognized at an advanced stage, which is why successful therapy is difficult and complex. The recurrence rate is very high depending on the genesis and also complicates the therapeutic success in the disease. The
mechanisms and correlations described here are therefore relevant and highlight potential targets for future therapeutic approaches and research points. The aspects described here are of importance and interest for the corresponding scientific field.
The article is understandable and well-constructed, and reflects an intensive examination of the current literature. The authors’ conclusions are consistently described and discussed by relevant literature. The essential signaling pathways and mechanisms have been summarized in an appealing figure.

R: We thank the Reviewer for his/her positive comments and appreciation of our manuscript.